# The Links between Parkinson’s Disease and Cancer

**DOI:** 10.3390/biomedicines8100416

**Published:** 2020-10-14

**Authors:** Maria Ejma, Natalia Madetko, Anna Brzecka, Konstanty Guranski, Piotr Alster, Marta Misiuk-Hojło, Siva G. Somasundaram, Cecil E. Kirkland, Gjumrakch Aliev

**Affiliations:** 1Department of Neurology, Wroclaw Medical University, Borowska 213, 50-556 Wrocław, Poland; mejma@interia.pl (M.E.); natalia.madetko@gmail.com (N.M.); konstanty.guranski@umed.wroc.pl (K.G.); 2Department of Pulmonology and Lung Oncology, Wroclaw Medical University, Grabiszyńska 105, 53-439 Wroclaw, Poland; anna.brzecka@umed.wroc.pl; 3Department of Neurology, Medical University of Warsaw, Kondratowicza 8, 03-242 Warszawa, Poland; piotr.alster@wum.edu.pl; 4Department of Ophthalmology, Wroclaw Medical University, Borowska 213, 50-556 Wroclaw, Poland; misiuk55@wp.pl; 5Department of Biological Sciences, Salem University, Salem, WV 26426, USA; siva.somasundaram@salemu.edu (S.G.S.); EKirkland@salemu.edu (C.E.K.); 6Sechenov First Moscow State Medical University (Sechenov University), St. Trubetskaya, 8, bld. 2, 119991 Moscow, Russia; 7Research Institute of Human Morphology, Russian Academy of Medical Science, Street Tsyurupa 3, 117418 Moscow, Russia; 8Institute of Physiologically Active Compounds, Russian Academy of Sciences, Chernogolovka, 142432 Moscow Region, Russia; 9GALLY International Research Institute, 7733 Louis Pasteur Drive, #330, San Antonio, TX 78229, USA

**Keywords:** Parkinson’s disease, cancer, oxidative stress, mitophagy, α-synuclein, PARK, PINK, DJ-1, mutations

## Abstract

Epidemiologic studies indicate a decreased incidence of most cancer types in Parkinson’s disease (PD) patients. However, some neoplasms are associated with a higher risk of occurrence in PD patients. Both pathologies share some common biological pathways. Although the etiologies of PD and cancer are multifactorial, some factors associated with PD, such as α-synuclein aggregation; mutations of PINK1, PARKIN, and DJ-1; mitochondrial dysfunction; and oxidative stress can also be involved in cancer proliferation or cancer suppression. The main protein associated with PD, i.e., α-synuclein, can be involved in some types of neoplastic formations. On the other hand, however, its downregulation has been found in the other cancers. PINK1 can act as oncogenic or a tumor suppressor. PARKIN dysfunction may lead to some cancers’ growth, and its expression may be associated with some tumors’ suppression. DJ-1 mutation is involved in PD pathogenesis, but its increased expression was found in some neoplasms, such as melanoma or breast, lung, colorectal, uterine, hepatocellular, and nasopharyngeal cancers. Both mitochondrial dysfunction and oxidative stress are involved in PD and cancer development. The aim of this review is to summarize the possible associations between PD and carcinogenesis.

## 1. Introduction

Parkinson’s disease (PD) is a progressive neurodegenerative disorder classified as an α-synucleinopathy. Clinically, it is mainly associated with movement disorders; however, in the course of the disease, many other symptoms can be observed. PD was first described in 1817 by James Parkinson [1], and it has remained in the scope of researchers’ interest ever since. Despite over 200 years of history, the exact mechanism and cause of this disease is still unknown.

The accumulation of α-synuclein plays a causal role in PD; notwithstanding, it cannot explain the selective pattern of neurodegeneration observed in the course of the disease [2]. High cytoplasmic concentrations of dopamine and calcium ions in cells forming the pars compacta of the substantia nigra (SN) and locus coeruleus may negatively affect accumulated α-synuclein. Its mutual interactions can increase the cytotoxic potential of these substances, enhancing selective vulnerability, leading to the neurodegeneration pattern characteristic for PD [3].

About 10% of PD cases are associated with mutations in α-synuclein, leucine rich repeat kinase 2 (LRRK2), DJ-1, PTEN-induced kinase 1 (PINK1), PARKIN, and several other proteins’ genes [4]. Hitherto, 26 loci associated with the risk of PD have been determined—inter alia, many PARK genes, for example, PARK1 and 4, encoding α-synuclein; PARK2, encoding parkin; PINK1, encoding PTEN-induced kinase 1 (PARK6); DJ-1 (PARK7); LRRK2 (PARK8); and ATP13A2 (PARK9). However, in the majority of patients, a genetic background cannot be found, and these cases are referred to as idiopathic PD.

Mutations in LRRK2 and the genes coding α-synuclein cause autosomal dominant hereditary PD, whereas mutations in DJ-1, PARKIN, PINK1, and ATP13A2 cause autosomal recessive variants. Many of these proteins are involved in mitochondrial or lysosomal functioning [5], and patients with PD present mitochondrial deficits and autophagy pathway impairment. These data suggest a possible role of mitochondrial and lysosomal dysfunction in PD’s pathogenesis. Similar anomalies are often described in the context of carcinogenesis [6].

Among the mechanisms collectively responsible for PD and neoplasm pathogenesis, mitochondrial dysfunction, oxidative stress, DNA damage, abnormalities in mitosis-stimulating signals, inflammatory factors, and cell cycle activation anomalies are listed [7]. Some biochemical substances, such as α-synuclein, which is one of the PD markers, may have a stimulating effect on malignant cells [8].

The aim of the study is to gather the data on the associations between PD and cancer and underline the possible common biological pathways shared by these two seemingly opposite pathologic processes.

## 2. Epidemiologic Links between Parkinson’s Disease and Cancer 

In many studies, the morbidity associated with majority of neoplasms is lower among patients with PD than in the general population. However, melanoma, brain tumors, thyroid gland cancer, and breast cancer were found to be related with higher risks in PD patients [9,10,11,12,13,14,15,16,17,18]. Additionally, an increased rate of patients affected by kidney and uterus cancers among PD patients was observed [19]. On the other hand, Peretz et al. [20] did not find differences in any-type cancer incidence among PD patients compared with the general population. This study was performed among 7125 PD patients in Israel (i.e., in a population with a frequent genetic basis for PD), and in 1301 of them (18.3%), de novo cancer diagnosis was established in a period encompassing 6.6 years before and 4 years after the beginning of PD treatment (in 72% of patients, before anti-Parkinsonian treatment). The other important finding of this study was that—similarly to in the general population—the incidence of any-type cancer in PD patients was higher in men than in women. However, there were significant differences in the incidence of some cancer types, such as a decreased incidence of lung cancer and colorectal cancer as compared with in the general population, with no significant differences in the occurrence of the other specific-localization cancer types.

In the other large-scale studies of PD patients, in Switzerland, there were decreased standardized mortality ratios in PD patients for lung cancer and liver cancer and increased for melanoma/skin cancer, breast cancer, and prostate cancer [21], and in Sweden, an increased risk of melanoma was found [22]. In a meta-analysis of 50 observational studies, it was estimated that among patients with PD, the morbidity related to malignant neoplasms was reduced by 17%, especially in the context of lung, bladder, bowel, prostate, and uterus cancers and blood malignancies [23]. Another meta-analysis revealed a decreased overall risk for cancer in PD patients, especially after excluding skin cancers [16]. In England, in a study encompassing 219,194 PD patients, a decreased overall risk for developing cancers was found, with an increased risk for breast cancer and melanoma [19]. In Denmark, in 14,088 PD patients, there was an increased risk for lung cancer, laryngeal cancer, and urinary bladder cancer and an increased risk for melanoma/skin cancer [24]. In Taiwan, in 62,023 PD patients, an increased risk of any cancer, including an increased risk of lung cancer and colorectal cancer, was found [25]. In a meta-analysis of 16 studies reporting prostate cancer and breast cancer, no association of these cancer types with PD was found [26]. A meta-analysis of the studies focused on the association of brain tumors with PD showed a higher incidence of central nervous system (CNS) malignant and non-malignant neoplasms in patients with PD [27]. The abovementioned authors, Peretz et al. [20], explained the differences in the incidence of cancer in different populations of PD patients by ethnic factors, cigarette smoking, alpha-synuclein deposits, and microbiome changes in the gastrointestinal tract, as well as by different time windows for the observations.

Some authors considered the temporal relationship between the appearance of cancer and PD, pointing out the differences in the risk of developing cancer depending on the period before or after the diagnosis of PD [13,28]. Many studies have shown a reduced incidence of cancer before the diagnosis of PD [28,29,30]. Similar analyses and differences also concerned the race and ethnicity of patients [13,14,31,32]. However, such differences were not always present; for example, in patients with PD and brain tumors, no relationship was found regarding gender and ethnicity [27], and in the Taiwanese population, despite a higher adjusted risk factor for melanoma development, it did not reach statistical significance [32].

The relationship between melanoma and PD and the high rate of the co-occurrence of PD and melanoma were noted in the 1970s [12]. The cause of that interest was the implementation of the treatment of PD with levodopa (L-DOPA). By that time, there was a widespread fear that this therapy could cause melanoma, as L-DOPA is a substrate in melanin synthesis. An iatrogenic relationship between melanoma and L-DOPA treatment has not been proven [33,34,35]. Currently, the more frequent occurrence of melanoma in patients with idiopathic PD (IPD) is beyond doubt [19,28,31,36,37]. Additionally, patients with melanoma have a higher risk of developing PD [38,39]. It has been observed that a family history of melanoma and lighter hair and skin color cause a higher risk of developing PD, and having a first degree relative with one disease carries a significantly increased risk of developing another. Although there is a link between PD and melanoma, the etiology of this relationship is still elusive. Both PD and melanoma are probably multidimensional diseases involving genetic and environmental risk factors. A potential factor linking these diseases may be environmental (e.g., exposure to pesticides, not smoking, and not consuming alcohol or caffeine) [40,41,42,43,44] or genetic. In families of patients with melanoma, a more frequent occurrence of PD [45] was observed; in patients with red hair color and MC1R polymorphism p.R151C, there was a more frequent occurrence of melanoma and PD [46]. It is highly likely that genetic factors reduce or increase the susceptibility of PD patients to malignant neoplasms. This is supported by reports of an increased risk of various cancers among patients with LRRK2 mutations [10,47,48]. It has been found that the LRRK2 G2019S mutation was associated with an increased risk of skin and breast cancer and the R1441C/G mutation, with the risk of colon cancer and hematological cancers. Comparing patients with IPD and patients with LRRK2-PD, the latter had a significantly increased risk of breast cancer, hormone-related cancers, and non-skin cancer [49]. In a study comparing patients with IPD, patients with LRRK2-PD, and persons from a control group, the patients with LRRK2-PD had a 4.6-fold increased risk of leukemia compared to the IPD patients and to the control group [10]. Additionally, mutations of other PARK genes found in familial forms of PD can increase the risk of developing cancer. In the case of PARK1 and PARK4 genes, this applies to lung, intestine, prostate, ovary, melanoma, and non-Hodgkin’s lymphomas [50,51,52,53]; in the case of PARK2, to glioma, lung cancer, ovarian cancer, kidney cancer, pancreatic cancer and melanoma [34,54,55,56]; in the case of PARK6, to glioma and ovarian cancer [57]; in the case of PARK7, to breast, lung, pancreatic, stomach, and prostate cancer [53,58,59]. Different PARK protein expression has been shown depending on tumor stages [53]. The less frequent occurrence of some tumors in PD, e.g., lung tumors, may result from the interaction between PARKIN and the p21 protein [60]. On the other hand, the higher incidence of breast cancer in PD may be associated with the lack of PARKIN protein expression in cancer cells (68%) or the rare methylation of the PARKIN promoter, which reduces PARKIN expression [61]. Some PD genes, including PARK1/4, PINK1, and PARK9 (ATP13A2), have already been detected to be associated with brain tumors, indicating that the same mutations can lead to pathological changes in both PD and brain tumors [27,50]. Cigarette smoking, which is a known risk factor for many malignant neoplasms, has also been considered as the cause of the negative relationship between cancers and PD [17]. There is a negative relationship between PD and smoking. Current smokers have a 60% reduced risk of developing PD, and former smokers have a 20% reduced risk [62]. It has been suggested that the explanation for the lower incidence of cancer among PD patients is a higher percentage of non-smokers or former smokers among those who develop PD compared to the general population. However, although this may explain the significant difference in the occurrence of smoking-related neoplasms, this does not explain the reduction in non-smoking-related cancer cases, and a lower risk of PD in patients has also been demonstrated for cancers not related to smoking [17,19,31].

It was attempted to explain the positive relationship between PD and brain tumors as a result of more frequent magnetic resonance imaging examinations in PD patients and the possibility of including in the analysis Parkinsonism secondary to the CNS proliferative process [27]. Some biochemical substances, such as α-synuclein, which belongs to PD’s markers, may have a stimulating effect on cancer cells [8]. α-synuclein (SNCA) decreases the activity of tyrosine hydroxylase, which reduces the production of dopamine and melanin [63]. Interaction between SCNA and tyrosinase may show up more frequently among patients with PD and dopamine deficiencies. The fibrillary forms of α-synuclein in PD may cause deviations of the tyrosine transformation process related to melanogenesis. According to this hypothesis, these disturbances may predispose PD patients to melanoma.

Other studies postulate a link between dopamine receptors (DR) and the risk of malignancies. DR polymorphisms have been shown to be associated with the risk of colorectal cancer, non-small-cell lung cancer, and gastric cancer, and increased expression of dopamine D2 receptors (DR2) has been observed in gastric cancer, neuroendocrine tumors, glioma, and breast cancer [64,65,66,67,68]. D2R antagonists have been reported to have antitumor efficacy in cell cultures and animal models, in which they reduced tumor growth, induced autophagy, influenced lipid metabolism, and caused apoptosis [69]. Increased DR2 expression was prognostically beneficial in neuroendocrine tumors and gastric cancer. D4R expression may, in turn, reduce the survival time of patients with glioblastoma multiforme [70].

Genetic mutations or polymorphisms found in patients with PD and their associations with different malignancies are summarized in Table 1.

## 3. Role of PINK1, PARKIN, and DJ-1 in Mitochondria

There are some common pathogenetic factors shared by PD autosomal recessive Parkinsonism and some types of cancer, such as the proteins PARKIN, PINK1 (PTEN induced putative kinase-1), and DJ-1 (Parkinsonism associated deglycase). These proteins play important roles on the subcellular level, participating in mitochondrial morphology, homeostasis, and function [71,72,73,74,75,76].

PINK1 can be located on both the outer mitochondrial membrane (OMM) and inner mitochondrial membrane (IMM), depending on the membrane potential. In polarized mitochondria, PINK1 is imported into the mitochondrial intermembrane space, where it is degraded by PARL (presenilin-associated rhomboid-like) proteases. A low level of PINK1 prevents the mitophagy of healthy mitochondria. By contrast, mitochondrial depolarization deactivates proteasomal degradation, leading to the accumulation of PINK1 in the OMM [76]. The PINK1 recruits PARKIN, which, after phosphorylation, is responsible for the ubiquitinylation of selected proteins on the mitochondrial surface, and the resulting ubiquitin chains are a signal for further degradation in the ubiquitin proteasome system. The PINK1/PARKIN pathway is activated by depolarization or mitochondrial damage [77]. Normal mitochondria continuously import PINK1 from the cytosol and degrade it partially in the matrix. PINK1 residues are exported for further degradation in the proteasome. When the mitochondrion is damaged, it loses the membrane potential necessary for the proper replacement of the components. In this situation, PINK1 is “trapped” in the OMM and begins to phosphorylate available proteins. PARKIN attaches ubiquitin to various proteins anchored in the OMM, thus marking the organelle as useless/harmful, intended for elimination and recycling. PINK1 additionally phosphorylates ubiquitin, activates PARKIN, and provides a strong signal for the phagophore to surround the mitochondrion. Mitochondrial homeostasis is essential for cell energy balance [78,79,80].

PINK1/PARKIN signaling is particularly important for the activity of dopaminergic neurons [81]. For not-completely-understood reasons, in the sporadic form of PD, damaged mitochondria begin to accumulate and generate free oxygen radicals that cause progressive degeneration and, finally, the death of neurons [76].

The PINK1/PARKIN pathway activation in mitochondria is summarized in the Figure 1.

## 4. PINK1, Parkinson’s Disease, and Cancer

PINK1 is a serine-threonine mitochondrial protein kinase with a mass of 63kDa. It consists of 581 amino acids. It can act as an oncogenic as well as a tumor suppressor gene. In its structure, it contains an N-terminal mitochondrial targeting sequence and C-terminal autoregulatory domain located towards the cytoplasm [82]. Physiologically, PINK1 levels are quite low because it is rapidly degraded. PINK1 is a vital neuroprotective protein designed to prevent mitochondrial damage and cellular apoptosis as a response to stress factors [83]. It stabilizes the potential of the mitochondrial membrane and stops the release of apoptogenic factors. It also participates in the processes of reactive oxygen species (ROS) production and oxidative phosphorylation. Research on PINK1 revealed that its many functions go beyond neuroprotection; this protein is involved in various diseases, including cancer [84]. PINK1 deficiency has also been shown to impair the plasticity of the striatum and hippocampus, which may result in neurodegenerative changes and cognitive impairment in PD [85]. PINK-1 and PI3-kinase/Akt signaling are interrelated and regulate each others’ activity [57]. More than 100 mutations in the PINK1 gene causing a loss of function in PINK-1 have been associated with autosomal recessive PD with early onset [86]. This protein is crucial in the regulation of the cell cycle and thus possibly favors cancer growth. On the other hand, however, cell cycle changes induced by the PINK1 deletion may result in chromosomal aberrations and lead to the development of cancer depending on the type of the cells [57]. Physiologically, PINK1 deficiency leads to altered mitochondrial calcium-buffering capacity [87].

PINK1 may be involved in some lung diseases, including lung cancer. The expression of this protein has been shown to be lower in the lungs of patients with idiopathic pulmonary fibrosis compared to in healthy individuals [88], while non-small-cell lung cancer (NSCLC) cells have shown higher levels [89]. In an experimental study on NSCLC, the cell depletion of PINK1 resulted in the cells’ decreased proliferation and increased mortality [90]. In lung adenocarcinoma, high PINK1 expression was significantly associated with chemo-resistance and was a poor prognostic factor, which was not confirmed for squamous cell lung cancer [91]. 

An experimental study revealed that the loss of PINK1 resulted in increased proliferation of glioma cells, reduced oxygen consumption, and increased glycolysis [92]. Hypoxia induced factor-1α has been stabilized in mouse neurons, fibroblasts, and astrocytes. A change in the metabolism of human astrocytes has also been observed. It has been proven that the loss of expression of PINK1 in neurons could induce cell death. This protein is significantly reduced in gliomas and medulloblastomas. By contrast, PINK1 re-expression inhibited ROS, glioma cell growth, and oxygen glycolysis. Patients with glioblastoma expressing the PINK1 protein have shown better survival. The role of PINK1 as a tumor suppressor gene was also confirmed in ovarian and breast cancer [92,93].

## 5. PARKIN, Parkinson’s Disease, and Cancer

PARKIN is a protein with a molecular weight of approx. 52 kDa, composed of 465 amino acids. The PARK2 gene, located in a highly unstable region on chromosome 6Q25-27, encodes it. Parkin has the ability of polyubiquitination and the regulation of cellular processes [94]. Through autoubiquitination, it can regulate its own activity [95].

It is crucial for the proper functioning of the ubiquitin–proteasome system and is involved in the regulation of the degradation of proteins participating in the process of apoptosis [96,97]. PARKIN is usually instanced in the cytoplasm of cells but is transferred to the mitochondria when they are damaged [98]; its translocation to mitochondria depends on the expression of PINK1.

Studies conducted on animals (mice) have shown that the accumulation of dysfunctional mitochondria, accelerated production of mtDNA mutations, and lack of PARKIN cause the degeneration of dopaminergic neurons and movement disorders [76,99]. PARKIN dysfunction is the cause of the familial form of PD (autosomal recessive juvenile Parkinsonism) and is also a risk factor for its idiopathic form [100]. PARKIN dysfunction can also cause neuronal damage in other degenerative diseases, e.g., Alzheimer’s disease, amyotrophic lateral sclerosis, multiple sclerosis, or Huntington’s disease [101,102,103]. PARK2 mutations may reduce the ubiquitinating ability for substrates such as α-synuclein or synphilin-1, leading to their toxic accumulation in the brain [104,105]. PARKIN also has anti-inflammatory effects, as PARKIN-knockout mice have an increased susceptibility to inflammation-related degeneration [106]. A lack of PARKIN inhibits the differentiation of neural stem cells by JNK-dependent (JNK-c-Jun N-terminal kinase) p21 proteasomal degradation [107]. 

PARKIN has been shown to contribute to mitosis regulation, which suggests that the inhibition of its activity may significantly redound to the initiation of the tumor formation process [108,109,110]. PARKIN disorders lead to the accumulation of mitosis regulators—PLK-1 kinase, the proteins Aurora A and B, cyclin B, and cyclin E. Consequently, chromosomal segregation disorders, micronuclei, and bipolar division spindles occur. Disorders resulting from mutations in the PARK2 gene are detected in approximately 30% of cancers, including lung, liver, intestine, and brain cancers; mice lacking the gene encoding parkin are more susceptible to developing cancer [110,111,112,113,114]. The loss of heterozygosity observed on chromosome 6q25-q26 may contribute to the initiation or progression of cancer by inactivating or reducing the expression of the Parkin gene [115]. Some tumors have also been shown to have reduced PARKIN mRNA levels [116,117,118]. PARK2 promoter changes have also been associated with the development of cancer, e.g., leukemia and renal clear-cell carcinoma [117,119]. Experimental studies have shown that PARKIN-deficient mice are susceptible to hepatocarcinogenesis [120]. PARKIN deficiency caused a change in hepatic gene expression profiles, increasing hepatocyte proliferation and their resistance to apoptosis, which led to the development of a liver tumor, partly by increasing the regulation of endogenous folistatin. By contrast, the ectopic expression of PARKIN in breast cancer cells with PARKIN deficiency reduced their proliferation rate both in vitro and in vivo, and also reduced the ability of these cells to migrate [121]. The accumulation of cyclin E and mitosis impairment have been demonstrated in colon and lung cancer cells and glioblastoma multiforme together with PARKIN dysfunction [110]. Much lower PARKIN expression in tumors with lymph node metastases has been reported in pancreatic, nasopharyngeal, and clear-cell kidney carcinomas compared to in tumors without the dissemination of cancer cells [117,122,123]. 

To date, PARKIN’s suppressive mechanism in cancer development has not been fully explained. It can be suspected that this protein may limit tumor growth by inhibiting the cell cycle or by regulating abnormal mitosis [124,125]. It is possible that it can also induce cancer cell apoptosis [126]. For example, it has been shown that PARKIN induces TNF receptor-associated factor (TRAF) 2 and TRAF6 proteolysis and, by that, favors the inhibition of nuclear factor-kappa-B (NF-κB), which affects apoptosis [118]. Lee et al. [127] reported that PARKIN expression restored TNF-α-induced apoptosis in HeLa cells, a human cervical cancer cell line. In their study, increased cell death was caused by the activation of the apoptotic pathway by reducing the expression of survivin, a caspase-inhibiting protein. PARKIN’s other anti-cancer activity is associated with the inhibition of the excessive glycolysis that typically occurs in cancer [128]. Research shows that PARKIN is a target for p53, as it mediates p53’s function in modulating molecular energy economy.

Participation in the necroptosis process may be another path for PARKIN’s influence on cancer [129]. Necroptosis is considered as a type of programmed cell death involving a cascade of molecular events leading to cell necrosis, controlled by its genome and dependent on RIPK3 (receptor-interacting serine-threonine kinase) [130]. In some cancer cells, necroptosis disorders and defects of important process effectors have been shown [131,132,133]. Based on their research, Lee et al. [106] suggested that the AMPK (AMP-activated protein kinase)/Parkin/RIPK3 pathway is a vital regulatory mechanism for necroptosis and inflammation-induced tumorigenesis. They showed that PARKIN deficiency exacerbates inflammation and inflammation-associated tumorigenesis, which indicates the protective role of PARKIN in carcinogenesis. A very important mechanism of tumor suppression is PARKIN regulation of mitophagy. Abnormalities in mitophagy may increase glycolysis and ROS production, and may promote cancer growth and metastasis.

All these observations indicate that PARKIN is actually a tumor suppressor, while its mutations may damage its function that inhibits carcinogenesis [116]. 

## 6. DJ-1, Parkinson’s Disease, and Cancer

DJ-1 is a multifunctional protein encoded by the PARK7 gene, containing 189 amino acids, located mainly in the cytoplasm but also found in the nucleus and mitochondria. It participates in cell survival, apoptosis, transcription regulation, and oxidative stress phenomena (it increases mitochondrial activity and protects neurons from death) [134,135,136]. DJ-1 is a chaperone that stabilizes the respiratory chain I complex and protects mitochondria from oxidative stress. DJ-1 integrates with the classic PINK1/PARKIN pathway [137,138]. It partially fulfills its antioxidant function by regulating the transsulfuration pathway, being a regulator of ferroptosis [139]. Ferroptosis is a type of programmed iron-dependent cell death characterized by the accumulation of lipid peroxides, which is genetically and biochemically different from other forms of regulated cell death, such as apoptosis [140,141]. DJ-1-mutated neuronal cells are thought to undergo high levels of ferroptosis and may therefore be involved in the mechanisms of early recessive PD. The overexpression of DJ-1 also reduces α-synuclein dimerization, while mutant forms of DJ-1 interfere with this process [142]. Direct interactions between DJ-1 and α-synuclein are the base neuroprotective mechanism. Family mutations in DJ-1 can contribute to PD by interfering with these interactions. Under stress, this molecule translocates from the cytosol to mitochondria, where it prevents the aggregation of α-synuclein.

DJ-1 was initially identified as an oncogene capable of transforming cells alone or cooperating with other oncogenes such as H-Ras and c-Myc143. It has been shown that this protein can promote cancer cell survival and their proliferation and metastasis formation by activating the Akt/mTOR, MEK/ERK, NF-κB, and HIFα pathways and through activating the PLAGL2/Wnt/BMP4 axis signaling pathways [143,144] DJ-1 may antagonize the PTEN tumor suppressor to inhibit PTEN gene activity. Similarly, it can work by repressing p53 and the JNK and ASK1 signaling pathways [145]. Increased DJ-1 expression has been found in various neoplasms such as melanoma or breast, lung, colorectal, uterine, hepatocellular, or nasopharyngeal carcinomas [71,146,147,148]. High levels of DJ-1 were significantly correlated with metastasis and worse prognosis in some cancers such as NSCLC, as well as in endometrial, pancreatic, esophageal, colorectal, and cervical cancers [143,144]. The overexpression of DJ-1 promoted the invasion, migration, and proliferation of colorectal cancer cells in vitro and in vivo [146]. Han et al. [149] suggested that DJ-1 might be a potential biomarker important for the early diagnosis and monitoring of lung cancer metastases. A high expression of DJ-1 in breast cancer cells was followed by increased HER3 signaling and promoted cancer cell proliferation in vitro and tumor growth in vivo [150]. Additionally, in medulloblastoma, a high expression of DJ-1 was associated with increased proliferation, undifferentiated tumors, high p-Akt expression, and a high MIB-1 index [151]. The high p-Akt expression was associated with tumor metastasis stage.

DJ-1 has also been identified as a negative regulator of ferroptosis in cancer cells, and the suppression of DJ-1 has been shown to promote ferroptotic cell death [139]. The downregulation of DJ-1 significantly suppressed the cell proliferation, migration, and invasion of papillary thyroid cancer cells, possibly by the DJ-1/PTEN/PI3K/Akt signaling pathway [152]. By contrast, the reduced expression of DJ-1 promoted inflammation and the apoptosis of intestinal epithelial cells by over-activating the p53 signaling pathway [153].

## 7. Cancer and α-Synuclein

SNCA (α-synuclein) is a protein monomer with a molecular weight of about 14 kDa. It consists of 140 amino acids. SNCA is encoded in 4q21. This protein is common in the CNS. Physiologically, SNCA is involved in the formation of synaptic plasticity, and the regulation of the transport and storage of dopamine in neuroendocrine vesicles. It is also associated with regulating dopaminergic transmission. SNCA may also be involved in pro- and antiapoptotic processes. It was also noted that SNCA aggregates, and diffuse accumulation may be observed secondarily to aging in neurologically healthy patients. SNCA is among the most relevant proteins involved in the etiopathogenesis of certain neurodegenerative diseases, e.g., PD. SNCA mutations lead to improper protein folding, aggregation, and the creation of insoluble protein deposits—Lewy bodies. Aggregation is associated with oligomer formation, causing neurotoxic effects and causing cell death in PD [154]. Exposure to oligomers is related to an increase in glutamatergic synaptic transmission, which prevents further potentiation [155]. Examination based on animals proved that exposure to SNCA may also be related to the loss of synaptic endings, the accumulation of dopamine in the cytosol, and disturbances in dopaminergic transmission [156]. Recently growing interest is associated with the prion-like spreading of SNCA fibrils. This may contribute to pathogenesis in PD [157,158,159].

Various studies suggest that the accumulation of SNCA is involved in carcinogenesis. Its expression was proved in melanoma, breast cancer, and ovarian cancer [160,161]. On the other hand, SNCA expression was reduced in lung adenocarcinoma cells [162]. Downregulation of SNCA correlated with decreased overall survival. In the most recent experimental studies with the human neuroblastoma SH-SY5Y cell line, it has been proved that the effect of increased expression of SNCA depends on its stage [163]. SNCA overexpression resulted in inducing increased cell vulnerability to oxidative stress. The effect was more significant at moderate increments of SNCA. Increased levels of this protein induced tumorigenicity in this cancer cell line. An immunoreactivity of α-synuclein can also be observed among brain tumors with neuronal differentiation and in schwannomas [50]. Li et al. [164] introduced SNCA as a novel diagnostic biomarker for medulloblastoma and proved that SNCA might inhibit cancer growth through apoptosis induction. The authors of the study stressed the role of epigenetic mechanisms, which play a vital role in the regulation of SNCA expression in medulloblastoma tumors. The hypermethylation of DNA was not found as relevant in medulloblastoma tumors. The methylation of the SNCA promotor was observed in lymphoma [51]. 

## 8. Mitochondrial Impairment in Parkinson’s Disease and Cancer

Mitochondria are vital cell organelles necessary for proper function. They participate in many processes, such as energy production (the synthesis of adenosine triphosphate, ATP), the generation of ROS, cellular metabolism, tumorigenesis, calcium homeostasis, and programmed cell death. Having many functions, they are structures susceptible to damage [165]. An increased number of dysfunctional mitochondria may participate in the pathogenesis of many diseases including cancer, diabetes, anemia, and neurological disorders—acute (e.g., ischemic stroke and mechanical trauma) as well as chronic neurodegeneration (e.g., Alzheimer’s disease, PD, Huntington’s disease, or amyotrophic lateral sclerosis) [166,167,168]. The malfunction of mitochondria is mainly manifested in an energy failure of the affected tissue, a decrease in the activity of electron transport chain complexes, the overproduction of free radicals, disorders of cellular Ca^2+^ ion homeostasis, the release of pro-apoptotic factors, and disorders of mitochondrial biogenesis. One of the main reasons for mitochondrial dysfunction is oxidative stress. Damaged mitochondria can be a signal for cell death, inflammation, or aging. The accumulation of damaged mitochondria primarily contributes to cell aging, which is most likely the result of the accumulation of ROS, inducing mutations in mtDNA.

Cancer is one of the diseases resulting from incorrect signaling in cellular systems related to cell survival and death [57]. The accumulation of damaged mitochondria occurs during tumor progression, hence the opinion that mitophagy, which is a type of selective autophagy consisting of mitochondrial degradation, is of key importance in the pathogenesis of cancer diseases, as well as neurodegenerative diseases [166].

Autophagy is an integral self-degradation process for high-molecular components of the cytoplasm, especially proteins with a long half-life and whole organelles [169]. It has an important role in cell survival, is responsible for maintaining intracellular homeostasis, and controls important physiological functions. It gains special significance in conditions of cellular stress, in situations of cell damage [170]. There are three forms of autophagy: microautophagy, macroautophagy, and autophagy dependent on chaperones. Autophagy can be a non-selective or highly selective process when specific structures are degraded (e.g., mitochondria—mitophagy; ribosomes—ribophagy; or bacteria and viruses—xenophagy). During mitophagy, the degradation of mitochondria is preceded by a loss of mitochondrial membrane potential and mitochondrial fragmentation. This process is important during the normal development and differentiation of certain cells (e.g., erythrocytes and reticulocytes). It maintains a balance between organelle biogenesis, protein synthesis, and the degradation of cellular components. Mitophagy is activated in response to hypoxia and mitochondrial damage [82]. The inefficient removal of damaged mitochondria is a hallmark of cell aging [171], and disorders in the quality control of mitochondria in neurons are considered to be the cause of many neurodegenerative diseases [172,173].

Autophagy is a multi-stage process, regulated via a group of genes called autophagy-related genes [174]. Multiple observations and experiments show that malfunctioning autophagy is the cause of many diseases, including neurodegenerative disorders, metabolic diseases, myopathy, obesity, cardiovascular diseases, and cancer [170,175,176]. A common feature of the diseases with a loss of autophagy is the accumulation of damaged organelles and dysfunctional proteins, which are useless material that inhibit the proper functioning of cells. Under conditions of impaired autophagy, the consequences of the presence of abnormally formed proteins are particularly well visible. In PD, it is α-synuclein; in Alzheimer’s disease, these are neurofibrillary tangles (tauopathy—hyperphosphorylation inactivating the microtubule-stabilizing tau protein); and in Huntington’s disease, they are polyglutamine-containing proteins (polyQ-containing proteins). Recent research indicates the role of microglia and selective autophagy (synucleinphagy) in the removal of α-synuclein released by neurons [177]. According to experimental studies conducted on mice, neuronal α-synuclein activates microglia, which then absorb α-synuclein into autophagosomes. This process takes place via the signaling pathway TLR4–NF-κB–p62 (microglial Toll-like receptor 4, which induces the transcriptional upregulation of p62/SQSTM1 through NF-κB). Interference with microglial autophagy favors the clustering of misfolded α-synuclein, leading to the degeneration of the dopaminergic neurons.

The deregulation of autophagy is currently considered to be one of the characteristic vital features contributing to cancer development. Autophagy plays a dual role in cancer; it can suppress the cancer process or protect cancer cells [178,179,180]. In neoplasms, autophagy can be inhibited as well as induced. For example, inhibiting autophagy and increasing the survival of cancer cells may be a consequence of the incorrect activation of the PI3K–Akt–mTOR signaling pathway [181]. This activation may be a result of the loss of the tumor transformation suppressor PTEN and the TSC1/TSC2 complex, the amplification or mutation of genes encoding class I PI3K kinases, the overexpression of PKB/Akt, and exposure to carcinogens, as well as an increased activity of tyrosine kinase receptors. The effects of such processes may also lead to the induction of protein translation, cell growth, and proliferation. The p53 protein, encoded by the TP53 gene, located on chromosome 17, also participates in the regulation of autophagy processes. This protein has the properties of a tumor transformation suppressor and can play a dual role in the regulation of autophagy. Depending on whether the p53 protein is present in the cell nucleus or cytoplasm, it can initiate or inhibit autophagy [182]. It has also been shown that p53 deficiency or mutated p53 variants that accumulate in the cytoplasm of cancer cells allow the activation of autophagy [183].

In the early stages of a neoplasm, autophagy acts as an anti-metastatic process by reducing cancer necrosis and inflammatory reactions [184]. It also reduces the invasion and migration of cancer cells from their places of origin. However, in advanced stages of tumor dissemination, autophagy favors metastasis formation and cancer cell survival. The knockdown of autophagy-related genes, such as Beclin 1 and LC3, inhibits proliferation, migration, and invasion and lead to apoptosis in breast cancer [185]. Melanoma cells show high levels of autophagy despite a reduction in the Beclin1 and LC3 genes [186,187].

## 9. Oxidative Stress, Parkinson’s Disease, and Cancer

Oxidative stress can be defined as an imbalance in the production and inactivation of ROS in favor of the free radicals resulting in cellular signaling disturbances or damage [188]. The idea of oxidative stress was first described in 1985 [189] and has remained scientifically noteworthy ever since.

### 9.1. Oxidative Stress in Parkinson’s Disease

PD can be characterized as a chronic, slowly progressive neurodegenerative movement disorder. Its main feature is the degeneration of dopaminergic neurons in the SN, which, at a certain level, becomes clinically noticeable [190]. Taking into consideration the clinical course of the disease with the gradual progression of its symptoms, it is likely that, on the cellular level, the pathological process responsible for neurodegeneration is persistent. Genetic pathologies found in some PD patients suggest a possible role of mitochondrial dysfunction and, caused by that, oxidative stress.

Free radicals are molecules with at least one unpaired electron incorporated in their orbitals [191]. This feature is responsible for the high reactivity of these molecules. The superoxide anion is considered as the most important and primary ROS in humans, as it is highly reactive and, due to its interactions, produces secondary ROS [192]. Uncontrolled oxidation leads to the mutilation of DNA, lipids, and proteins, which is noticed in PD patients [193,194]. Free radicals in humans can be generated during aerobic respiration in mitochondria (not-fully-reduced oxygen leakage), by phagocytes in the course of infection, in peroxisomes, by other types of oxidative metabolism, or by behavioral factors, such as cigarette smoking or drinking coffee. It is possible that uncontrolled oxidation makes cells more and more damage-prone. Interestingly, in the SN of healthy individuals, the concentration of oxidized proteins was found to be twice that in the caudate, putamen, and frontal cortex, indicating that the susceptibility of the SN to oxidative stress may contribute to the selective neuronal degeneration [195].

In the context of PD and oxidative stress, iron involvement is very often discussed in the literature. Iron is classified in the group of “essential” metals, as it participates in cell homeostasis sustainment and regulates and catalyzes biochemical reactions conducted by many enzymes, receptors, or transporters. However, an excess of iron may lead to oxidative stress and damage to neurons [196]. Iron is important for the proper functioning of many CNS enzymes such as tyrosine hydroxylase, glutamate decarboxylase, monoamine oxidase A and B, or tryptophan hydroxylase. It has been proven that abnormal iron accumulation in the CNS is characteristic for PD [197]. In PD, this pathological iron accumulation is accompanied by an inflammatory process, which is not observed in the aging brain [198].

In biological environments, iron can be found in two oxidative states: Fe^2+^ and Fe^3+^—these two forms constantly transform into each other in redox reactions. A popular theory about the pathogenesis of PD includes the formation of ROS during a Fenton reaction involving the iron inbuilt in neuromelanin (NM). This theory was based on the fact that postmortem examinations revealed higher concentrations of iron in the SN in patients with PD compared to in healthy controls [199,200]. However, many articles undermine the veracity of this thesis. It remains unknown whether NM is synthesized in an enzymatic reaction or is a product of dopamine derivative oxidation. Some authors suggest that NM is made of a plethora of catecholamine residues, which are not retained in synaptic vesicles, which might be a neuroprotective mechanism [201]. 

Dopamine and its metabolites located in neurons’ cytoplasm undergo oxidation, which results in the production of ROS [202]. During this process, H_2_O_2_ is formed. This molecule participates in Fenton reactions with free, unbound iron ion Fe^3+^, in which hydroxyl radicals are formed. Patients with PD have higher concentrations of unbound iron in the SN compared to healthy controls [203,204,205]; however, the general amount of iron remains similar among these groups. Based on these facts, it can be supposed that the decline of NM’s ability to bind iron may lead to increased ROS production and, in the end, cell death. It is also possible that the iron participating in Fenton reactions is bound to NM, as it was proven that the free ion concentration required for this reaction is too high for cell survival [206]. However, in that case, the free radicals formed as a result of Fenton reactions would be neutralized by NM before they could spread through the cytoplasm. 

Another theory considering PD’s pathogenesis involves NM and calcium ions. NM is responsible for Ca^2+^-buffering capacity, and when the concentration of NM declines, a cell becomes more vulnerable to fluctuations in the calcium ion level. That theory is a Ca^2+^ hypothesis of PD [207]. The author suggested that NM’s loss in dopaminergic neurons was a first phase of cell death. Neurons with lower concentrations of NM lose the ability to protect themselves from changes in cytosolic calcium ions levels, which may lead to apoptosis. 

The pathogenesis of PD is a complex issue. Oxidative stress is widely discussed as a possible mechanism also present in physiological aging [208]. Another important component might be a neuroinflammatory process as a result of cell death and the deposition of abnormal proteins [209]. Some authors describe a specific phenotype for dopaminergic neurons in the SN, i.e., the senescence-associated secretory phenotype (SASP) [210], which is a result of both oxidative stress and neuroinflammation. Cellular senescence is an anti-cancer mechanism that enables cell proliferation and replication to be stopped. SASP is present in the astrocytes of patients with PD and can damage neighboring cells by the bursty secretion of proinflammatory chemokines, proteases, or growth factors. This phenomenon may prevent cancer development; however, in the long term, it has harmful effects on neurons in the SN that could lead to neurodegeneration. 

### 9.2. Oxidative Stress in Cancer

Oxidative stress is one of the primary causes linked with inflammation and carcinogenesis [211]. Oxidative stress is a DNA damaging factor in the induction of carcinogenesis in nicotine stimulation [212]. ROS of endogenous and exogenous origin are associated with oxidative stress, which may eventually cause deviations in DNA repair and cell proliferation [213]. ROS are produced in various reactions such as those associated with NADPH oxidase, xanthine oxidase, uncoupled endothelial nitric oxide synthase (eNOS), arachidonic acid, and metabolic enzymes such as the cytochrome P450 enzymes, lipoxygenase, and cyclooxygenase [214]. The production of ROS can be additionally increased by high psychological stress and dietary disorders, among which can be mentioned high fat intake, which is widely observed [215].

An excessive amount of ROS usually activates apoptotic signaling pathways and initiates cell death. However, in some cases, when cell cycle regulation and signaling are ineffective, carcinogenesis is observed.

Changes determined by oxidative stress influence transcription factors and second messengers [213]. Oxidative stress, when accompanied by chronic inflammatory processes, may lead to mutagenesis and neoplasm development [216]. The cascade of pathological reactions results in a possible increase in cancer transformation [213]. Oxidative stress accompanied by the hyperactivation of insulin-like growth factors and adipokines is interpreted as more deregulated among obese women [217]. Contemporarily, the cell response to oxidative stress and metabolism are associated with regulation controlled by the p53 protein [218]. A surplus of ROS activates the p53 protein, which is a tumor suppressor. This protein activates mechanisms of DNA repair and temporally arrests the cell cycle. If this DNA repair process is ineffective, p53 promotes the apoptosis of the damaged cell. However, when oxidative stress is persistent and cellular antioxidative mechanisms are overtaxed, tumor suppressors are inactivated and the cell cycle is promoted by oncogenes such as EKR, Akt, and c-MYC [219]. During that process, a damaged cell is able to proliferate, and the number of mutations increases, as the cell cycle is no longer supervised and DNA material is already damaged. ROS lead to carcinogenesis not only by directly interacting with DNA but also by the products of membrane lipid peroxidation such as malonic dialdehyde or 4-hydroxynonenal [219], which interact with DNA. It is suggested that ROS, as a marker of oxidative stress, are vital for cancer neoangiogenesis [220]. Oxidative stress’ level of importance in carcinogenesis depends on the neoplasm nature and its specific signaling receptors.

An important factor participating in the balance between oxidative stress and carcinogenesis may be nuclear factor erythroid 2-related factor (NRF2). This transcription factor is a detector of oxidative stress in a cell. Physiologically, it is located in the cytoplasm in a complex with Keap1. During oxidative stress, NRF2 dissociates from Keap1, translocates to the nucleus, and promotes the transcription of genes coding for antioxidative enzymes and proteins [219]. This protective function occurs when NRF2 is activated occasionally. However, when this factor is active permanently, it is associated with cancer growth and drug resistance [221]. It prevents the apoptosis of the cancer cell and enhances its proliferation [222]. 

Correlations between oxidative stress and carcinogenesis may be used for therapeutic aims in the future. For example, the effect of oxidative stress on carcinogenesis according to various studies shows significant changes during the use of metformin [223,224]. The anti-cancer property of metformin is not fully examined. However, some reports indicate the induction of autophagy and apoptosis among patients with colon cancer, one of the most common entities in oncology [225]. Metformin pharmacotherapy is related with cell cycle arrest in the G0–G1 phase [226]. The stimulation of oxidative stress at the initiation and progression stages of cancer may be interpreted as a protective factor [227]. A profitable impact of treatment using metformin was brought up in the treatment of colon and breast cancer [225,226]. This shows that antioxidants widely recommended as favorable for general health may, in certain periods, enhance carcinogenesis [227]. On the other hand, antioxidants such as these supplements with phytochemicals in plant-based foods have a protective role in the phases preceding carcinogenesis. In one of the studies, the authors evaluated the role of curcumin, epigallocatechin gallate, resveratrol, phenethyl isothiocyanate, sulforaphane, hesperidin, quercetin, and 2’-hydroxyflavanone. Preventive roles are based on the protection of physiological intracellular molecular mechanisms [228]. This shows why ROS-elevating and eliminating treatments are introduced in various therapies [229].

In general, oxidative stress and its impact on carcinogenesis are, in many ways, controversial. Antioxidants, though causing a reduction in possible damage to DNA, may, during the initial phase of carcinogenesis, prevent the apoptosis or necrosis of cancer cells. The mechanism of inducing apoptosis as an effect of oxidative stress is used in various therapies. Development in the knowledge concerning oxidative stress shows how its certain features may be used in modern oncology.

## 10. Comments

PD, being an example of a neurodegenerative disease, and cancer—a disease of uncontrolled cellular multiplication—might be considered as two biologically opposite processes. The question arises whether this opposition translates into clinical observations. Epidemiological data equivocally show associations between PD and cancer. Although most reports, including analyses based on studies of thousands of patients, indicate an inverse association between PD and cancer, this is not true in all cancer types. There are also strong environmental influences on the incidence of both diseases, mostly smoking. The basis of the observed inverse relation between smoking and PD has not been elucidated yet, and direct associations between smoking and some cancer types is undisputable. However, some of the cancers not related to smoking may also be inversely associated with PD.

Despite the deceptive opposition of these pathologies, where PD gradually leads to the death of neurons and cancer is characterized by rapid proliferation, both disorders share biological pathways. The role of the factors typical for PD, such as α-synuclein or DR, although found to be also related with some cancer types, is still far from being elucidated in cancer development in PD patients. There is increasing evidence indicating vital roles in the pathophysiology of both diseases played by mitochondrial dysfunction, the production of ROS, oxidative stress, DNA damage, cell cycle abnormalities, and impaired mitophagy (Figure 2).

The most promising explanation of positive or negative associations between PD and some cancer types is common mutations occurring in both diseases. Molecular studies have shown that the proteins coded by genes associated with PD participate in mitochondrial activities, and mutations in these proteins affect both neurodegeneration and tumorigenic processes. Proteins such as α-synuclein, PARKIN, PINK1, and DJ-1, acting together, play an important role in various cellular functions, influencing antioxidative responses, and the morphology and function of mitochondria, especially in tissues sensitive to hypoxia.

Expanding knowledge on the etiopathogenesis of PD and cancer will allow the development of novel diagnostic methods that may have clinical applications. An explanation of the potential mechanisms involved in these pathologies and their collective dependencies could form the basis of effective targeted therapies that modify and improve the course and prognosis in both diseases.

## Figures and Tables

**Figure 1 biomedicines-08-00416-f001:**
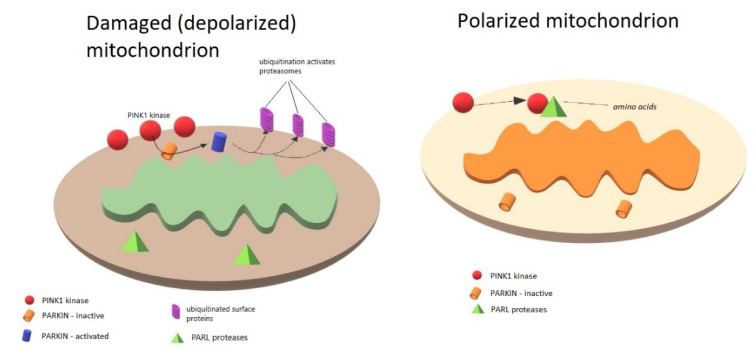
PINK1/PARKIN pathway activation in polarized mitochondria. Depolarization of mitochondria/mitochondrial damage activates PARKIN-dependent mitophagy by induction of PINK-1 kinase in the outer mitochondrial membrane (OMM). Loss of membrane potential necessary for component replacement is responsible for increased concentration of PINK1 in the OMM. PINK1 stabilizes and phosphorylates Parkin, which, when activated, is responsible for ubiquitinylation of proteins on mitochondrial surface, which is a signal for proteasomes to degrade the mitochondrion. In polarized mitochondria, PINK1 is continuously being imported into the mitochondrial intermembrane space, where it is degraded by PARL proteases. Low levels of PINK1 prevent mitophagy of healthy mitochondria.

**Figure 2 biomedicines-08-00416-f002:**
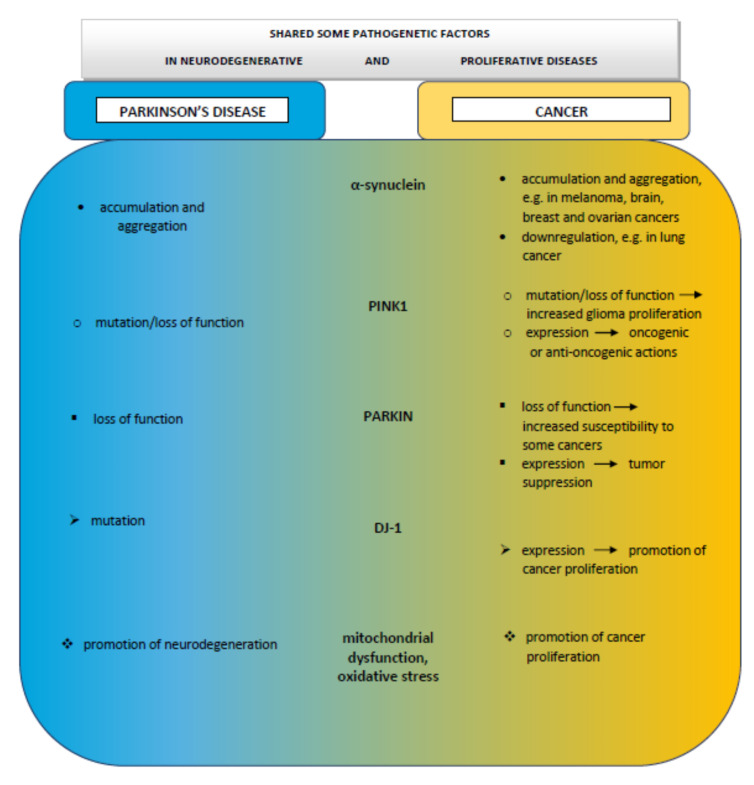
Schematic drawing of possible association between Parkinson’s disease and cancer.

**Table 1 biomedicines-08-00416-t001:** Genetic mutations and polymorphisms associated with different malignancies and found in Parkinson’s disease (PD) patients.

Mutation/Polymorphism	Type of Neoplasm
MC1R polymorphism p.R151C	melanoma
LRRK2 G2019S mutation	skin cancer, breast cancer
R1441C/G mutation	colon cancer, hematological malignancies
LRRK2-PD	non-skin cancer, hormone-related cancers, breast cancer, leukemia
PARK1, PARK4	Lung, intestine, prostate, and ovarian cancers; melanoma; non-Hodgkin’s lymphoma
PARK2	lung, ovary, kidney, and pancreatic cancers; glioma; melanoma
PARK6	glioma, ovarian cancer
PARK7	breast, lung, pancreatic, stomach, and prostate cancers
PARK1/4, PINK1, PARK9 (ATP13A2)	brain tumors
DR polymorphisms	gastric, colorectal, and non-small-cell lung cancers;
increased expression of dopamine D2 receptors	gastric cancer; breast cancer; neuroendocrine tumors; glioma

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
