# Peer review of "The Links between Parkinson’s Disease and Cancer"

_biomedicines, 2020, doi:10.3390/biomedicines8100416_

Round 1
Reviewer 1 Report
Overall, this review is well written. This reviewer would recommend minor point to improve this manuscript.
1: The author should improve figure 1 and 2 to make leaders easily catching content.
Author Response
Dear Reviewer 1
On behalf of my coauthors, please accept my sincere thanks and gratitude for careful perusal and critical review of our manuscript
Comments:
The author should improve figure 1 and 2 to make leaders easily catching content.
Response: Thank you for this remark. Both Figures have been changed, revised, and replaced in the revised manuscript file.
Reviewer 2 Report
it is helpful for readers understand the relationship between PD and cancer to summerize the knowledge of these two popular diseases. However, this review is hard to say it is a well written in English. Except the writing, some major revison need to be done.
1. In the line 33-35, please add some detail about Peretz et al research. Analysis the difference between this study and other studies.
2. In the section 3. Role of PINK1, PARKIN and DJ-1 in Parkinson disease and cancer, I don’t think authors explain it clearly.
3. Please re-organize the figure 2.
4. Review is definitly not just summerizing. Please give some valuable comments about the reported relationship between PD and cancer.
Author Response
Dear Reviewer 2
On behalf of my coauthors, please accept my sincere thanks and gratitude for careful perusal and critical review of our manuscript.
Comments:
- In the line 33-35, please add some detail about Peretz et al research. Analysis the difference between this study and other studies.
Response: Thank you for this suggestion.
1.a. Accordingly some details of Peretz et al study were added:
Peretz et al. [20] did not find the differences in any-type cancer incidence among PD patients compared with general population. This study was performed among 7 125 PD patients in Israel (i.e. in the population of frequent genetic basis of PD) and in 1 301 of them (18.3%) de novo cancer diagnosis was established in a period encompassing 6.6 years before and 4 years after beginning of PD treatment (in 72% of patients – before antiparkinsonian treatment). The other important findings of this study was that – similarly as in the general population – the incidence of any-type cancer in PD patients was higher in men than in women. However, there were significant differences in the incidence of some cancer types, as decreased incidence of lung cancer and colorectal cancer as compared with general population were found with no significant differences in the occurrence of the other specific-localization cancer types.
1.b: Accordingly, analysis of the differences between Peretz study and other studies was enlarged:
In the other large-scale studies of PD patients, in Switzerland there was decreased standardized mortality ratios in PD patients for lung cancer and liver cancer and increased – for melanoma/skin cancer, breast cancer and prostate cancer [21] and in Sweden: increased risk of melanoma was found [22]. In a meta-analysis of 50 observational studies it was estimated that among patients with PD, the morbidity related to malignant neoplasms was reduced by 17%, especially in the context of lung, bladder, bowel, prostate, uterus cancers and blood malignancies [23]. Another meta-analysis revealed decreased overall risk for cancer in PD patients, especially after excluding skin cancers [16]. In England, the study encompassing 219 194 PD patients decreased overall risk for developing cancers in was found, with increased risk for breast cancer and melanoma [19]. In Denmark, in the 14 088 PD patients there was increased risk for lung cancer, laryngeal cancer and urinary bladder cancer and increased risk for melanoma/skin cancer [24]. In Taiwan, in 62 023 PD patients increased risk of any-cancer, including increased risk of lung cancer and colorectal cancer were found [25]. In the meta-analysis of 16 studies reporting prostate cancer and breast cancer no association of these cancer types with PD was found [26]. Meta-analysis of the studies focused on the association of brain tumors with PD showed a higher incidence of central nervous system (CNS) malignant and non-malignant neoplasms in patients with PD [27]. The above mentioned authors, Peretz et al. [20], explained the differences in the incidence of cancer in different populations of PD patients by ethnic factors, cigarette smoking, alpha-synuclein deposits and microbiome changes in the gastrointestinal tract, as well as by a different time window of observations.
- In the section 3. Role of PINK1, PARKIN and DJ-1 in Parkinson disease and cancer, I don’t think authors explain it clearly.
Response: Thank you for showing us necessity of improving the text in the section 3. We have modified the title (omitting “in Parkinson’s disease and cancer”), much shortened (removing unnecessary details), and corrected this part of the article. Now it will have the shape as follows.
- Role of PINK1, PARKIN and DJ-1 in mitochondria
There are some common pathogenetic factors shared by PD autosomal recessive parkinsonism and some types of cancer, such as the proteins PARKIN, PINK1 (PTEN induced putative kinase-1), and DJ-1 (Parkinsonism associated deglycase). These proteins play important role on the subcellular level, participating in mitochondrial morphology, homeostasis and function [71-76].
PINK1 can be located on both the outer mitochondrial membrane (OMM) and inner mitochondrial membrane (IMM), depending on the membrane potential. In polarized mitochondria, PINK1 is imported into the mitochondrial intermembrane space, where it is degraded by PARL (presenilin-associated rhomboid-like) proteases. Low level of PINK1 prevents mitophagy of healthy mitochondria. In contrast, mitochondrial depolarization deactivates proteasomal degradation, leading to the accumulation of PINK1 in the OMM [76]. The PINK1 recruits PARKIN, which, after phosphorylation, is responsible for the ubiquitinylation of selected proteins on the mitochondrial surface, and the resulting ubiquitin chains are a signal for further degradation in the ubiquitin proteasome system. The PINK1/PARKIN pathway is activated by depolarization or mitochondrial damage [77]. Normal mitochondria continuously import PINK1 from the cytosol and degrade it partially in the matrix. PINK1 residues are exported for further degradation in the proteasome. When the mitochondrion is damaged, it loses the membrane potential necessary for proper replacement of the components. In this situation, PINK1 is “trapped” in the OMM and begins to phosphorylate available proteins. PARKIN attaches ubiquitin to various proteins anchored in OMM, thus marking the organelle as useless/harmful, intended for elimination and recycling. PINK1 additionally phosphorylates ubiquitin, activates PARKIN and provides a strong signal for the phagophore to surround the mitochondrion. Mitochondrial homeostasis is essential for cells energy balance [78, 79, 80].
PINK1/PARKIN signaling is particularly important for the activity of dopaminergic neurons [81]. For not completely understood reasons, in the sporadic form of PD damaged mitochondria begin to accumulate, generate free oxygen radicals that cause progressive degeneration, and finally the death of neurons [76].
3.Please re-organize the figure 2
Response: Thank you for indicating that the illustration is not clear enough. The figure 2 has been re-organized.
- Review is definitly not just summerizing. Please give some valuable comments about the reported relationship between PD and cancer
Response: Thank you for indicating that our “Final remarks” were not sufficient as authors’ comments. And accordingly, the final paragraph was enlarged.
- Comments
PD, being an example of neurodegenerative disease, and cancer – a disease of uncontrolled cellular multiplication might be considered as two biologically opposite processes. The question arises whether this opposition translates into clinical observations. Epidemiological data equivocally show the associations between PD and cancer. Although most reports, including analyses based on the studies of thousands of patients indicate inverse association between PD and cancer, this is not true in all cancer types. There are also strong environmental influences on the incidence of both diseases, mostly smoking. The basis of the observed inverse relation between smoking and PD has not been elucidated yet and direct association between smoking and some cancer types is undisputable. However, some of the cancers not related to smoking may be also inversely associated with PD.
Despite the deceptive opposition of these pathologies, where PD gradually leads to the death of neurons and cancer is characterized by rapid proliferation, both disorders share biological pathways. The role of the factors typical for PD, such as α-synuclein or DR, although found to be related also with some cancer types, is still far from being elucidated in cancer development in PD patients. There is increasing evidence indicating vital role in the pathophysiology of both diseases played by mitochondrial dysfunction, production of ROS, oxidative stress, DNA damage, cell cycle abnormalities and impaired mitophagy (Figure 2).
The most promising explanation of positive or negative associations between PD and some cancer types are common mutations occurring in both diseases. Molecular studies have shown that the proteins coded by genes associated with PD participate in mitochondrial activities and mutations in these proteins affect both neurodegeneration and tumorigenic processes. The proteins like α-synuclein, PARKIN, PINK1 and DJ-1 acting together play important role in various cellular functions, influencing anti-oxidative response, morphology and function of mitochondria, especially in the tissues sensitive to hypoxia.
Expanding knowledge on the etiopathogenesis of PD and cancer will allow development of novel diagnostic methods that may have clinical application. Explanation of potential mechanisms involved in these pathologies and their collective dependencies can pose the basics of effective targeted therapies that modify and improve the course and prognosis in both diseases.
Round 2
Reviewer 2 Report
I think the authors have addressed every point in the comments. Even there still a little bit wrong in style, I would like to suggest to accept it.